# Road extraction through Yangwang-1 nighttime light data: A case study in Wenzhou, China

Anfeng Zhu[1,2], Jun Hao[1,2], Xu Gang[1,2], Hao Zhang[1,2]*, Xiaoyu Long[3], Luyao Wang[4]

1 College of Artificial Intelligence, Zhejiang College of Security Technology, Wenzhou, China, 2 Wenzhou Key Laboratory of Natural Disaster Remote Sensing Monitoring and Early Warning, Wenzhou, China, 3 School of Public Administration, Zhongnan University of Economics and Law, Wuhan, China, 4 State Key Lab for Information Engineering in Surveying, Mapping, and Remote Sensing, Wuhan University, Wuhan, China

* 20096354@zjcst.edu.cn

**Data Availability Statement:** All data sources are avaliable and described in the paper.

**Funding:** This research was funded by Basic Social Development Science and Technology Project of Wenzhou Science and Technology Bureau (Grant

## Abstract

The extraction of roadways from remote sensing imagery constitutes a pivotal task, with far-reaching implications across diverse domains such as urban planning, management of transportation systems, emergency response initiatives, and environmental monitoring endeavors. Satellite images captured during daytime have customarily served as the primary resource for this extraction process. However, the emergence of Nighttime Light (NTL) remote sensing data introduces an innovative dimension to this arena. The exploration of NTL data for road extraction remains in its nascent stage, and this study seeks to bridge this gap. We present a refined U-Net model (CA U-Net) integrated with Cross-Attention Mechanisms, meticulously designed to extract roads from Yangwang-1 NTL images. This model incorporates several enhancements, thereby improving its proficiency in identifying and delineating road networks. Through extensive experimentation conducted in the urban landscape of Wenzhou City, the model delivers highly accurate results, achieving an F1 score of 84.46%. These outcomes significantly surpass the performance benchmarks set by Support Vector Machines (SVM) and the Optimal Threshold (OT) method. This promising development paves the way towards maximizing the utility of NTL data for comprehensive mapping and analysis of road networks. Furthermore, the findings underscore the potential of utilizing Yangwang-1 data as a reliable source for road extraction and reaffirm the viability of deploying deep learning frameworks for road extraction tasks utilizing NTL data.

## 1. Introduction

Road extraction from remote sensing images is a critical task in numerous applications such as urban planning, sustainable transportation management, emergency response, and environmental monitoring [1–3], which the cornerstone of a nation's sustainable economic and social development. Over the years, the development and progress of road extraction technology have been driven by the need for accurate and efficient road mapping and monitoring. Remote

NO.S20220017. The funding was received by first author Anfeng Zhu, and his role includes data collection, method design and writing of original manuscript.

**Competing interests:** The authors have declared that no competing interests exist.

sensing techniques, including satellite and aerial imagery, have emerged as valuable tools for this purpose, offering advantages such as extensive area coverage, up-to-date information, and reduced costs compared to traditional ground-based surveys [4–6].

The evolution of road extraction methods can be traced from early pixel-based approaches to more advanced object-based and deep learning techniques. Initially, pixel-based methods, such as thresholding and edge detection, were commonly used for road extraction tasks [7]. Despite the numerous benefits and potential of satellite imagery, road extraction from such data can pose several challenges. Current approaches predominantly depend on the expensive high-resolution satellite images and are also struggled with the complex nature of urban and rural environments, which involve varying road materials, occlusions by buildings and vegetation, shadow effects, and diverse terrain conditions. These ground features often occupy more than 80% pixels of one satellite images and their texture and spectral features will significantly cause impacts on the quality and accuracy of the extracted roads. These challenges often stem from the complexity and diversity of road features, as well as the varying quality and resolution of satellite imagery. Other environmental factors, such as shadows of ground objects, can also impact the effectiveness of road extraction algorithms [8]. Therefore, we believe it's crucial to utilize a new type of data sources rich in road information, while also being cost-effective in the extraction of road information.

While daytime satellite images have been widely used for road extraction, the use of nighttime light (NTL) remote sensing data offers a new perspective [9,10]. NTL data, derived from sensors like the Visible Infrared Imaging Radiometer Suite (VIIRS) onboard the Suomi National Polar-orbiting Partnership (NPP) satellite and the Operational Linescan System (OLS) onboard Defense Meteorological Satellite Program (DMSP) platforms, can reveal unique patterns of human activities and urbanization. Nighttime light (NTL) remote sensing data, as captured by sensors such as the Visible Infrared Imaging Radiometer Suite (VIIRS) and the Operational Linescan System (OLS), has been widely adopted in various fields due to its unique capability to depict human activities and urban development. It provides a comprehensive view of human settlements and activities on a global scale, making it a valuable resource for numerous applications [11–15]. In the field of urban studies, NTL data has been widely used to monitor urbanization patterns and rate. Researchers utilize NTL data to detect the spatial extent of urban areas and their temporal changes, providing critical insights into urban growth and sprawl [16,17]. The ability to capture urban development over time also makes NTL data a valuable tool for tracking and modeling urban dynamics. In socioeconomic studies, NTL data has been employed as a proxy for economic activity. Given that economic activity is often correlated with light emissions, NTL data has been used to estimate economic indicators such as GDP and energy consumption at various spatial scales, from local regions to countries. In population studies, NTL data has been used for population estimation and distribution. The spatial distribution of nighttime lights is often associated with human settlements, making it a useful data source for estimating population density and distribution. In disaster management, NTL data is used to assess the impacts of natural disasters. Changes in nighttime light intensity can reflect the degree of damage and recovery, providing valuable information for disaster response and recovery planning.

Despite the numerous successful applications of Night-Time Light (NTL) data in socioeconomic studies, its use in road extraction remains relatively unexplored. Only a handful of studies have endeavored to employ NTL images for road extraction tasks, achieving some noteworthy results. For instance, Wang [18] harnessed Luojia 1–01 NTL imagery to extract roads in Wuhan through shape optimization and a thresholding approach, attaining a final extraction accuracy of 83.2%. However, the resolution of Luojia 1–01 NTL data is relatively low. In contrast, the higher resolution (30 m) Sustainable Development Goals Satellite-1

(SDGAST) was utilized in Chang's [9] research, where a watershed segmentation approach was employed to extract roads in Beijing and Wuhan. This method yielded an extraction accuracy of 84.65% when compared to OpenStreetMap (OSM) road network data. Although these studies underscore the potential and feasibility of using NTL data for road extraction, they also highlight several inherent limitations and challenges. (1) The adoption of NTL data for ground object extraction tasks remains limited—a possible reason might be that the majority of NTL research focuses on the use of coarse-resolution NTL images such as VIIRS and DMSP data. This focus is largely due to their free accessibility and straightforward processing workflow. However, these coarse-resolution NTL datasets often fall short in detecting the edges of ground objects, such as roads, thereby posing significant challenges in road extraction tasks. The overglow effect—where light from urban areas spills over to adjacent regions—can result in the overestimation of road extents in these rough-resolution images [9]. Moreover, the relatively coarse spatial resolution of NTL data compared to high-resolution daytime imagery can limit the ability to capture detailed road features. That also makes the usage of NTL data mostly concentrated on socioeconomic aspects. Nowadays, more NTL satellites with higher resolution have been launched, benefitting from the technical advancement and the prospective application of NTL data in sustainable development goals. The emergency of these new NTL images gradually makes ground objection extraction possible with low cost. However, the study on these new generation NTL data is still lacking. (2) Some external data sources, like OSM data, taxi GPS data, and smartphone GPS data, are often combined with VHR imageries when performing road extraction [19–21], however, that rarely happens during the analysis of NTL data set. A very important reason is that the previous NTL data do not have the capacity to match with the external data sources due to low resolution. (3) The analysis method of current NTL data is also very simple–mainly threshold-based methods and morphology-based methods, which can be sufficient for macroscopical issues, like GDP estimation, population estimation and consumption potential prediction [22,23]. But these methods are not enough to solve the objective recognition tasks. Therefore, another challenge is to improve the current NTL analysis methods to adapt to the new possibilities of these new generation NTL images.

In recent years, deep learning models have shown great potential for extracting features and patterns from complex data, including remote sensing images [24–29]. These models, particularly Convolutional Neural Networks (CNNs) and their variants, have been successful in overcoming many limitations of traditional road extraction methods. The U-Net model, one of the most prominent deep learning architectures, has demonstrated promising results in various remote sensing applications, particularly in road extraction tasks. Nevertheless, the standard U-Net architecture has limitations when applied to road extraction, especially in complex urban and rural environments. Several studies have attempted to enhance the performance of deep learning models for road extraction by incorporating additional features and strategies. For instance, Chen, *et al.* [30] proposed a modified U-Net model that incorporates multi-scale contextual information and residual connections to improve the accuracy of road extraction. Wang, Peng, Li, Alexandropoulos, Yu, Ge and Xiang [6] combined U-Net with graph-based techniques to refine the road extraction results and better handle complex road network structures. Furthermore, Yang, *et al.* [31] proposed a deep learning model that combines U-Net with a conditional random field (CRF) to better delineate road boundaries and achieve more accurate road extraction results. Given these challenges, there is a need to develop robust and effective methods for road extraction from NTL remote sensing data. Deep learning techniques, particularly convolutional neural networks (CNNs), have shown great promise in handling complex image segmentation tasks, such as road extraction from high-resolution daytime satellite image. However, their potential in handling NTL data for road extraction is yet to be fully exploited.

In addressing these research gaps and amplifying the accuracy of road extraction via NTL data, this study attempts to prove the feasibility of employing the new generation NTL satellite —Yangwang-1—in road extraction endeavors. We derive an improved U-Net model (CA U-Net) combined with cross-attention mechanisms to extract information from NTL images and incorporating characteristics from OpenStreetMap (OSM). We hypothesize that the unique properties of NTL data, combined with the powerful feature learning capabilities of deep learning models, can lead to accurate and effective road extraction. We also compare the extraction results with traditional road extraction methods—Support Vector Machines (SVM) and Optimal Threshold (OT), which do not use OSM data in training process, and further find that the proposed method can archive better performance (accuracy) than traditional methods. The main contributions of our work can be concluded as bellow: (1) We tested the capacity of new generation NTL imagery—Yangwang-1, in road extraction tasks. To our knowledge, it's the first attempt in adopting 30m resolution NTL imagery in ground object extraction. (2) We firstly combine NTL data and the external data set—OSM data in the training process through a deep learning framework and cross-attention mechanisms. Our approach largely improves the road extraction accuracy of traditional threshold methods, which use NTL data only and the street network data (OSM) is just used for accuracy assessment. Our research can provide references for high-accuracy and low-cost road extraction through fine-resolution NTL imagery.

## 2. Materials and methods

### 2.1 Study area

Situated on the southeastern coast of China in the Zhejiang province, Wenzhou, renowned for its economic dynamism and entrepreneurial spirit, lies at approximately 27.99˚ N latitude and 120.69˚ E longitude, as shown in Fig 1(A). As a significant contributor to the Chinese private economy, Wenzhou is both economically important and culturally vibrant [32,33]. Spread over an area of 11,784 square kilometers, the city boasts a population of over 9 million people as of the 2020 census. The cityscape is diverse, encompassing both urban and rural environments with a mix of densely populated city centers, industrial areas, and more serene suburban and rural regions. The city is well-known for its unique landscapes, including mountains, rivers, islands, and coastal lines, making it a charming destination in the region. The city of Wenzhou has a well-developed transportation system, acting as a critical hub for both regional and national transportation. The road network is vast and extensive, with the total length of roads exceeding 4,300 kilometers. The city is connected by a comprehensive network of expressways and highways, providing effective connectivity within the city and to other major cities in the province and beyond. The intricate web of roads facilitates the transportation of goods and movement of people, thus playing a significant role in the city's economic activities.

### 2.2 Data Sources and preprocessing

On June 11, 2021, Origin Space—a space recourse utilization firm in China, launched the Yangwang-1 ("Look Up 1") satellite, which is a small optical space telescope that uses visible and ultraviolet observations to detect near-Earth asteroids [17]. However, this satellite can spot more than the asteroids. For example, it can capture footage of the aurora australis and meteors as they strike the Earth's atmosphere. Besides, it can collect the NTL images by tuning its camera to target the ground. The visible band of Yangwang-1 has a wavelength between 420 and 700 nm, which is different from that of VIIRS/DNB and Luojia-1. It collects nighttime imagery with a ground spatial resolution of 38 m at the nadir. The off-nadir image from Yangwang-1 was taken at 23:15 local time (UTC 13:15) on September 15, 2021, as shown in Fig 1(B).

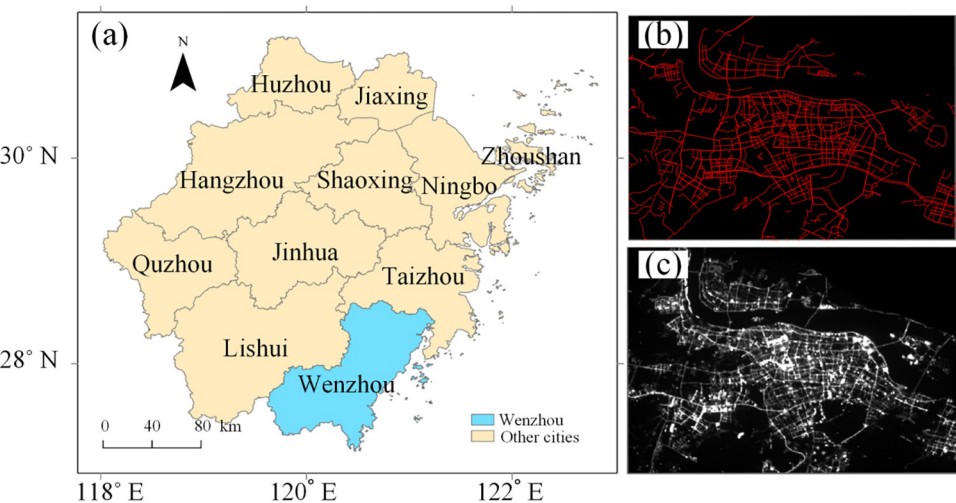

**Fig 1.** (a) Location of Wenzhou in Zhejiang Province. (b) Distribution of roads derived from Open StreetMap (OSM) in Wenzhou. (c) Yangwang-1 NTL data in Wenzhou.

The weather at the acquisition time was generally clear without clouds and heavy aerosols. The raw image is projected with the equatorial coordinate system. The pixel values are digital numbers (DNs) ranging from 0 to 4095 (i.e., 12 bits).

We also use roads derived from OpenStreetMap for model training and references, the distribution of which is shown in Fig 1(C). OpenStreetMap (OSM) represents a community-driven mapping initiative that provides geospatial data with global coverage, serving as a critical source for various academic, commercial, and humanitarian activities [34]. Roads in OSM contain a variety of attributes, like type of road (motorway, residential, footpath, etc.), its name, surface type, speed limit, number of lanes, and other relevant information. These tags are a powerful tool for describing features and are a key part of the OpenStreetMap's flexibility. While the quality and level of detail in OSM's data may vary depending on the geographical area, its dynamic nature allows for continuous updating and improvements through global contributions. OSM's road data has found significant application in machine learning tasks, such as road network extraction from satellite imagery, and broader contexts of urban planning and disaster management [35].

Following the acquisition of Yangwang-1 NTL and OSM data, a series of preparatory steps are necessitated prior to their deployment for road extraction. (1) Initially, both datasets are projected into the WGS-84 coordinate system, a pivotal step for preserving spatial integrity and ensuring alignment between them. (2) Subsequently, data registration is performed utilizing fifteen control points to ascertain the spatial congruity of these datasets. These control points are judiciously dispersed across the NTL imagery and predominantly selected from road junctions, given the paucity of other discernible ground objects within the NTL data. (3) Thereafter, pixels situated in water bodies and mountainous terrains are excised from the NTL data through spatial overlay with OSM land use data. (4) To mitigate adverse influences from background noise and the blooming effects inherent in NTL data, thresholds are established with a lower limit of 1 and an upper limit of 3325, the latter representing the maximal radiance value observed within the commercial center (pixels exhibiting 4095 radiance are identified as blooming effects). Consequently, pixels manifesting radiance exceeding 3325 are normalized to 3325. This thresholding strategy is devised to preserve all prospective road pixels whilst eliminating non-lit areas or noise, thus honing the focus on the urban roads of interest. (4)

Subsequently, the original roads are transmuted into raster data congruent with the resolution of NTL data (30m resolution), with pixels being registered with NTL data. Within the raster data, pixels are bifurcated into road pixels and non-road pixels. (5) The Yangwang-1 NTL data alongside the road raster data are then partitioned into 1462 pairs of image segments, each encompassing 32×32 pixels. (6) Finally, these images are stochastically apportioned into training samples comprising 1022 pairs of images, and a test set constituting 440 pairs of images, thereby priming the data for ensuing analytical endeavors.

The flowchart of this study is shown in Fig 2, in which three main steps are included: (1) Data Preprocessing. The Yangwang-1 NTL imagery and OSM data are projected into a uniform WGS-84 coordinate system, and undesirable pixels within the NTL imagery are expunged. Subsequently, OSM roads are transmuted into raster data, delineating between road pixels and non-road pixels. The NTL data and OSM raster data are then bifurcated into

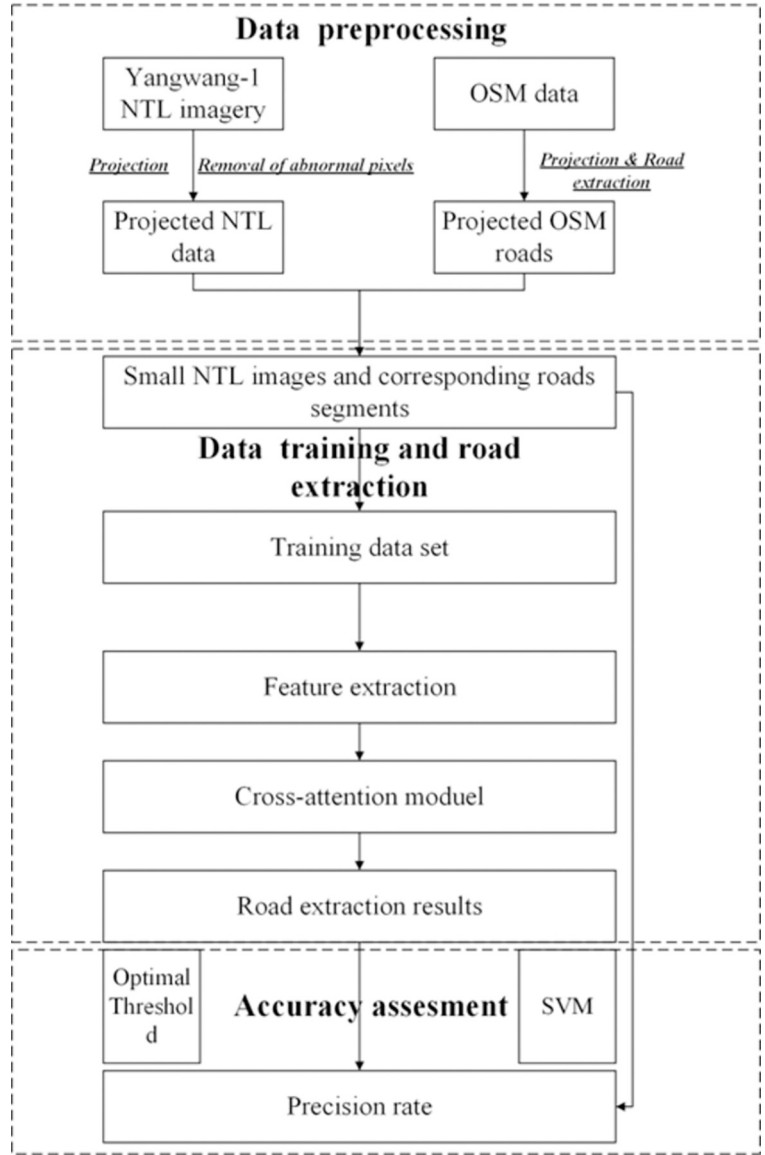

**Fig 2. Flowchart of the road extraction process.**

paired segments, primed for data training. (2) Data Training and Road Extraction. The segmented smaller NTL images and corresponding road segments are stochastically selected to constitute training and testing samples. The training dataset serves as the training ground for the CA U-net network, embodying three integral modules—feature extraction, cross-attention module, and road generation module. (3) Accuracy assessment. The roads generated through training samples are then compared with the testing samples derived from original OSM roads for accuracy evaluation. We further use two traditional methods–optimal threshold (OT) method and support vector machines (SVM) method to extract roads from the same training samples, and then compare their extraction accuracy with CA U-net model.

## 2.3. Support Vector Machines (SVM)

Support Vector Machines (SVM) is a robust machine learning algorithm renowned for its ability to handle high-dimensional data and non-linearly separable classes, as shown in Fig 3. Applying SVM to nighttime light (NTL) data for road extraction can provide a compelling mechanism to decode the intricate patterns associated with human settlement and transportation infrastructure. SVM operates by identifying an optimal hyperplane in a high-dimensional feature space that maximally separates different classes of data [36–38]. When applied to road extraction from NTL data, each data point can represent a pixel or a group of pixels in an image, with several associated features forming the high-dimensional space. These features can include the intensity of light, location attributes, and other derived features from the image.

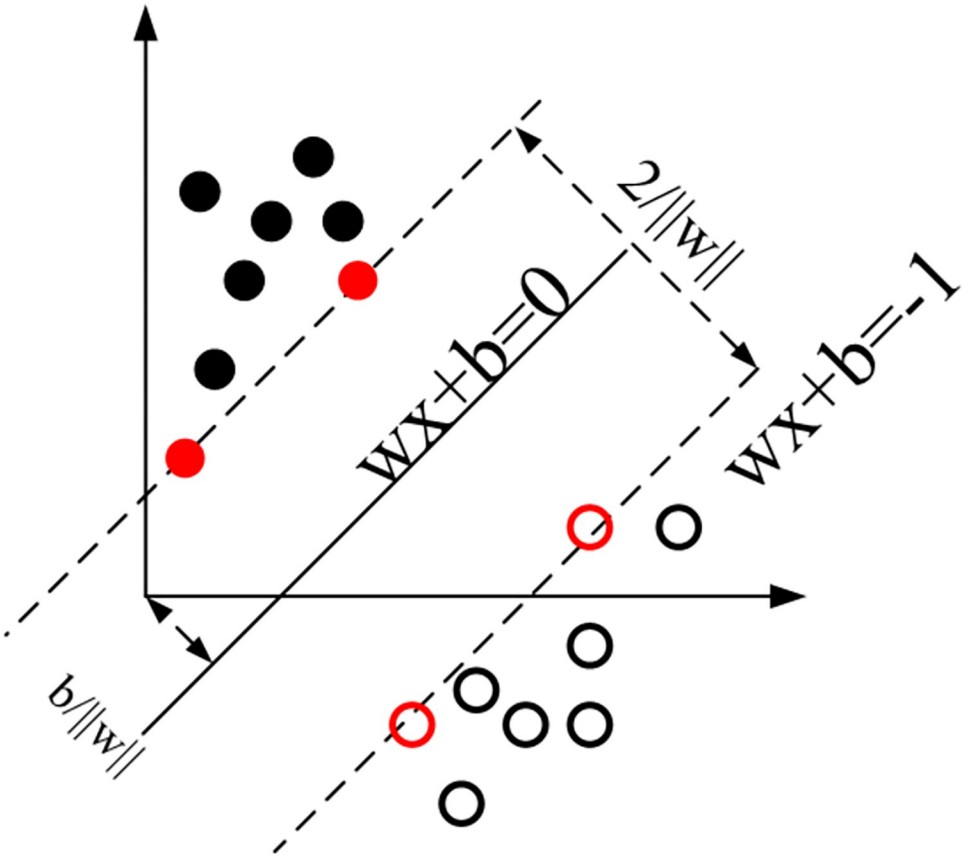

**Fig 3. Principle of the SVM algorithm.** The solid line represents the decision surface. Circles located in dotted lines are the support vectors.

The hyperplane in this context would aim to separate 'road' and 'non-road' classes. The hyperplane can be given as:

$$wx - b = 0, \qquad (1)$$

Where w is the weight vector, x denotes the input vector and b is the bias. However, it is often the case that the NTL data is not linearly separable due to various factors such as light blooming effects, sensor limitations, or atmospheric interference. SVM tackles this issue by implementing a mechanism known as the 'kernel trick', which projects the data into a higher-dimensional space where a separating hyperplane can be found. The equation with the kernel trick would be:

$$f(x) = B(0) + \sum ti \times K(x, x_i), \qquad (2)$$

Where $K(x, x_i)$ is the kernel function. It computes the dot product of the vectors in the higher-dimensional space. $ti$ are the Lagrange multipliers obtained by solving the dual problem. In the context of NTL data, the SVM could use a Gaussian Radial Basis Function (RBF) or polynomial kernel, among others, depending on the complexity and distribution of the data. Once the SVM model is trained, it can classify new unseen data based on its learning, thereby identifying roads from NTL data. As such, SVM can serve as a powerful tool for extracting road networks from NTL data, offering a cost-effective, scalable, and potentially more accurate method compared to traditional road extraction techniques. Moreover, SVM's flexibility allows it to integrate with other machine learning models or techniques, offering promising avenues for future research in road extraction from NTL data.

## 2.4 Optimal threshold

The 'optimal threshold' method offers a straightforward and efficient approach in image segmentation tasks such as road extraction from nighttime light (NTL) data. With its core concept hinging on setting varied intensity thresholds to distinguish between 'road' and 'non-road' areas based on light emission, this method lends itself to simplicity and flexibility [39]. The threshold works to either minimize the within-class variance or maximize the between-class variance of these classes [15,16]. For each threshold value, the extracted road length from NTL data is compared with a reliable source of actual road length, such as OpenStreetMap data, thus the optimal threshold would be the one rendering the closest match. However, while its simplicity and ease of application stand as significant benefits, the method does carry inherent limitations. It operates under an assumption of homogeneous distribution across classes, which might not align with real-world NTL data subject to variations in lighting conditions, sensor limitations, and atmospheric effects. Furthermore, the optimal threshold method can display high sensitivity to noise and outliers within the data, with a single misclassified pixel possessing the potential to significantly skew the threshold determination. Finally, it's important to note that this method overlooks the spatial or contextual information of pixels, which can be a critical factor in road extraction tasks.

In sum, the optimal threshold method provides an accessible and flexible approach for road extraction from NTL data, but careful consideration of its limitations is necessary. Combining this approach with other methods could potentially yield a more robust and accurate road extraction methodology.

## 2.5 Cross-attention U-Net framework

In this study, we designed an improved U-Net framework for road extraction incorporating a novel attention module. The U-Net framework, encompassing four encoder layers and four

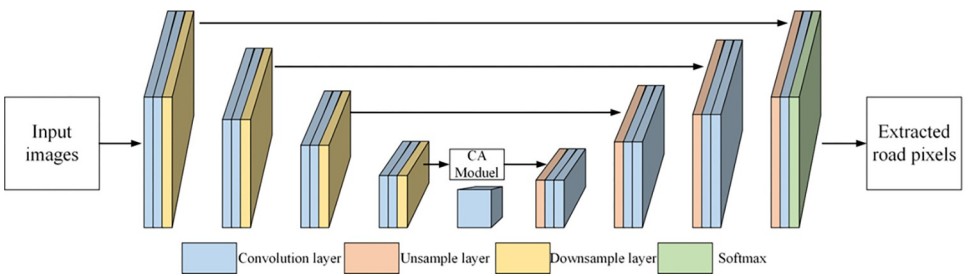

**Fig 4. The proposed CA U-Net framework for road extraction.**

decoder layers, accepts remote sensing image as input and processes it through the network to yield a binary image road network map [40,41], as shown in Fig 4. We adopted a Cross-Attention module proposed in [42], which is used for text-image retrieval, and combined the module with our U-Net framework. The details of the CA module are shown in Fig 5. In our CA U-Net framework, each encoder and decoder layer are embedded with residual dense connection blocks, facilitating the connection of feature information at different levels for efficient transmission of information flow. Moreover, the spatial attention block and the channel attention block are cascaded and jointly embedded into the U-Net framework, enhancing the utilization of spectral information while improving spatial details. Through the CA U-Net framework, the characteristics of roads pixels within NTL data can be learned and trained, which have not been conducted in previous research. We transfer the deep learning technologies dealing with daytime remote sensing images to the object detection tasks through NTL data and a cross-attention module, and firstly adopt the OSM roads in training process, which largely enhance the ability to capture road characteristics in NTL imagery.

The architecture of U-Net framework comprises two key components: an encoder layer for feature extraction, and a decoder layer for semantic segmentation. The encoder captures the

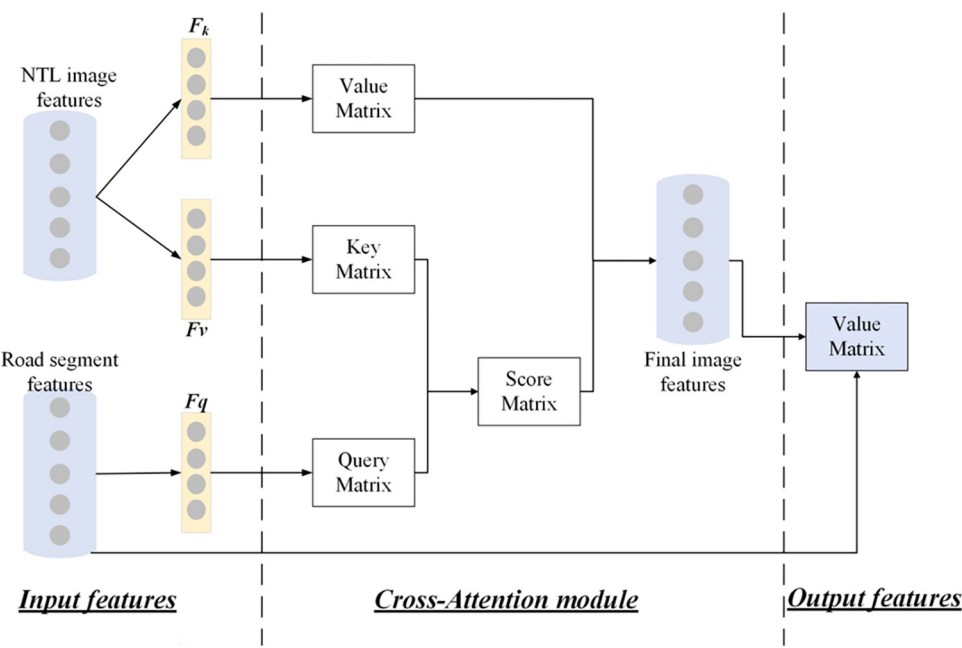

**Fig 5. The structure of the cross-attention module in U-Net framework.**

contextual information through a series of convolutional layers followed by 2x2 max-pooling operations for downsampling. As we move deeper into the encoder layers, we acquire more abstract and high-level semantic information, while the spatial dimensions reduce. The decoder layer decodes these feature maps, restores the spatial dimensions and resolution through a series of up-convolution operations, and locates the region of interest (ROI) in the image to obtain semantic segmentation results. A unique aspect of U-Net architecture is the incorporation of 'skip connections' from each encoder layer to the corresponding decoder layer, allowing for the fusion of low-level features with high-level semantic information. This contributes to precise localization and better preservation of details.

Adding to the U-Net architecture, attention mechanisms are used for recalibrating the weights of the feature maps, enabling the model to focus on specific, informative areas. Local attention mechanisms are employed for recalibrating the spatial information of remote sensing images. These mechanisms perform feature compression on the feature maps, reduce the dimensionality to two dimensions, and obtain a two-dimensional weight through a normalization function, thereby ensuring high-frequency information receives higher weight.

CAM is responsible for converging the regional-level image features with text features serving as a guiding factor. Initially, we ascertain the correlation weights between every region and the corresponding texts. Following this, a weighted summation leveraging these weights is executed, resulting in the synthesis of the overall semantic features inherent in the images. Subsequently, the similarities between the image and text are computed. In the end, we yield the matrix that encapsulates the similarities between the image and text. The CAM's comprehensive structure is visually depicted in Fig 5. The ensuing explanation elaborates on the specific computational process. Employing three fully connected layers, we map the region-level image features into both a key tensor and a value tensor. These tensors collectively hold the semantic information of the images. Simultaneously, the text features are transformed into a query matrix that encapsulates all the semantic information of the texts.

Use the three fully connected layers to map region-level image features into a key tensor and a value tensor containing all semantic information of images and text features into a query matrix containing all semantic information of texts.

$$M_k = F_k = (F, \theta_k), \tag{3}$$

$$M_v = F_v = (F, \theta_v), \tag{4}$$

$$M_q = F_q = (T', \theta_q), \tag{5}$$

Where $M_k$ and $M_v$ represent $n \times 1024 \times l$ tensors, in which n represents the number of images, 1024 represents the space dimension and l denotes the number of image areas. $M_q$ represents the number of road segments. Through the formulas, we can further calculate the cosine distance between each area in NTL image $S_{ij}$ and road segment $R_l$.

$$dis_{ij}^{(l)} = \frac{(S_{ij}^T, R_l)}{||S_{ij}^T|| \cdot ||R_l||}, \tag{6}$$

We then normalize the cosine distance through

$$Ndis_{ij}^{(l)} = \frac{e^{dis_{ij}}}{\sum e^{dis_{ij}}}, \tag{7}$$

According to the weight score, we can find the feature representation of the $i$th image fused with the $l$th piece of road segment information:

$$f_i^{(l)} = \sum\nolimits_{j=1}^{J} Ndis_{ij}^{(l)} v_{ij}, \qquad (8)$$

Where $v_{ij}$ denotes the $j$th area of image $v_i$. Therefore, we can then generate the similarity between the $i$th image and $l$th road segment:

$$Sim_{il} = \frac{f_i^{(l)T} t_j'}{||f_i^{(l)}|| \cdot ||t_j'||}, \qquad (9)$$

By employing a U-Net framework imbued with CAM, we are able to extract and leverage both global and local information, and focus on critical features with high precision, thereby achieving proficient road extraction from NTL data.

## 3. Experiments and results

### 3.1 Extraction results

The extracted roads by CA U-Net model, shown in Fig 6(B), seem to have best shape compared with OSM data (Fig 6(A)) with fewer fragmentations and lower overflow than SVM (Fig 6(C)) and OT (Fig 6(D)).

In an effort to conduct an in-depth comparison of road extraction results derived through various models, we selected three emblematic local areas within the urban expanse of Wenzhou City, as illustrated in Fig 7>(A)–7(C). Area (a) corresponds to Yaoxi Street in Wenzhou,

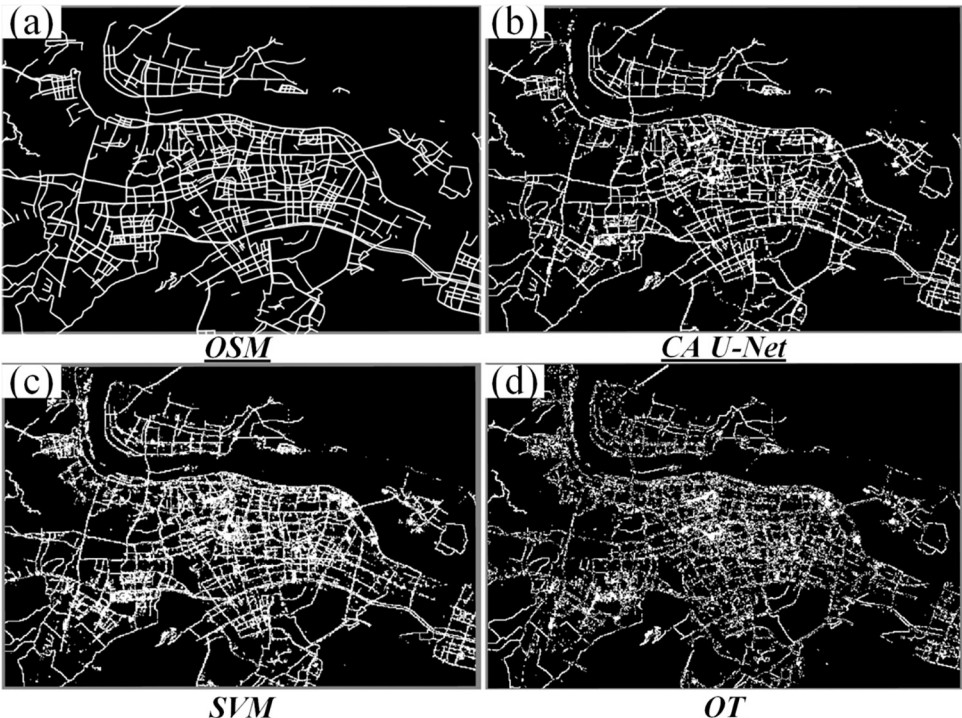

**Fig 6.** (a) The distribution of roads derived from OpenStreetMap. (b) Roads extracted by CA U-Net framework. (c) Roads extracted by SVM. (d) Roads extracted by OT.

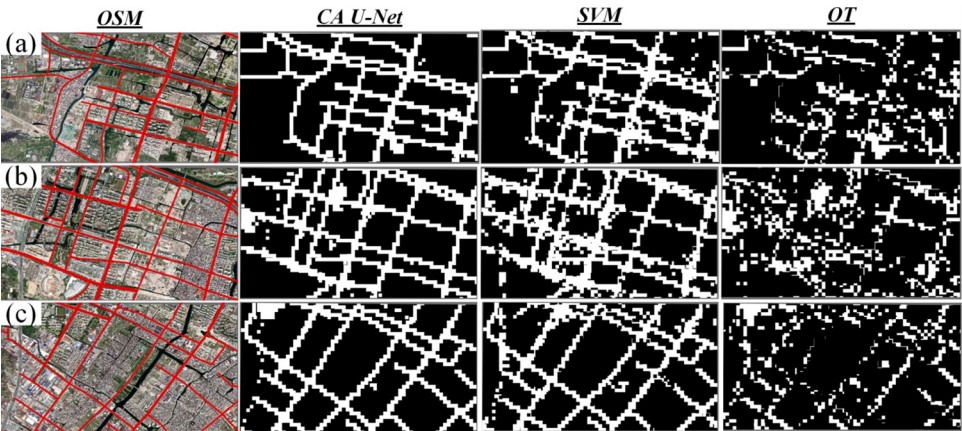

**Fig 7. The three representative places to show the extraction performance of three models.** (a) Yaoxi Street, a suburban area with 45% green land. (b) Yongzhong Street, located nearby the transition region of suburban region and urban region. (c) Adjacent part of Yongzhong Street and Shacheng Street, covering large amount of residential area and urban village area.

situated within the city's suburban periphery and encompassing approximately 45% of the region's green space. This includes the forests nestled in Yaoxi Mountain, Yaoxi Park, and agricultural land. Area (b) pertains to Yongzhong Street in Wenzhou, located near the transitional boundary between suburban and urban regions. The western portion of this area predominantly features high-rise buildings and expansive roads, while the eastern part is characterized by older, low-rise houses and footpaths. Area (c) demarcates the contiguous boundary of Yongzhong Street and Shacheng Street, ensconced along the Jinsha River's riverfront. This region incorporates a substantial portion of residential and urban village areas, comprising a complex network of roadways, including elevated roads, bridges, and expressways. Districts (b) and (c) typically exhibit a higher traffic density relative to district (a), attributed to their higher residential density. Upon comparing the road extraction results from these three areas, it is apparent that the roads extracted by the CA U-Net framework exhibit a greater congruity with the reference road network, in comparison to those extracted by SVM and OT. Through our comparative analysis, we observed that SVM could yield satisfactory extraction accuracy in suburban locales, such as roads (a) and (c). However, its ability to distinguish roads in bustling areas with extremely high light intensity seems to be notably inferior to that of CA U-Net, as evidenced in the left section of road (b). The performance of OT across all regions was found to be subpar, suggesting that it may not be the ideal choice for road extraction unless used in conjunction with other methods.

## 3.2. Accuracy verification

We employed measures such as precision rate, recall rate, and the F1 score, a harmonic mean of precision and recall, to evaluate the accuracy of road extraction. Drawing from references [4,9], precision rate signifies the proportion of correctly extracted sections (those overlaying with OSM data) to all extracted roads. In contrast, the recall rate represents the ratio of correctly extracted parts to all OSM roads. The F1 score is the harmonic mean of the precision and recall rates. We conducted our analysis predominantly utilizing the ArcGIS Pro 3.0.1 platform, capitalizing on its robust spatial analysis module and accompanying Python library. These tools not only facilitate comprehensive analysis but also allow for secondary development. Within the ArcGIS Pro deep learning framework, we integrated the cross-attention

**Table 1. F1 scores of CA U-Net, SVM, and OT models.**

| Model | Precision rate (%) | Recall rate (%) | F1 Score (%) |
|-------|-------------------|-----------------|--------------|
| CA U-Net | 87.51 | 81.62 | 84.46 |
| SVM | 74.26 | 70.95 | 72.56 |
| OT | 47.01 | 42.67 | 44.73 |

algorithm into the U-Net module layers. This integration was achieved using the secondary development tool, ArcPy. Subsequently, we employed this newly developed module to train a dataset comprising 1,022 labeled image pairs. The training procedure required approximately 28,376 seconds, averaging 27.76 seconds per image pair. Furthermore, the SVM algorithm was implemented via ArcPy within the ArcGIS Pro environment, consuming roughly 245 seconds for the entire categorization process. Lastly, we applied the OT method using the raster calculator feature in ArcGIS Pro, which required a mere 62 seconds upon selecting the appropriate thresholds.

As illustrated in Table 1, the F1 scores for CA U-Net, SVM, and OT stand at 84.46%, 72.56%, and 44.73%, respectively. These results highlight that both CA U-Net and SVM have F1 scores surpassing 70%, affirming the viability of Yangwang-1 NTL data as a data source for road extraction. Additionally, the substantially higher F1 score for BO-MWSA compared to SVM and OT, underscores the appropriateness of the proposed framework for road extraction from NTL data.

Typically, the nighttime lighting of roads is brighter than that of buildings, yet in densely built-up areas, building illumination can exceed that of roads. This difference in illumination intensity can lead to misclassification in road extraction.

## 4. Discussion

### 4.1 Advantages of Yangwang 1 images compared with other NTL platforms

Nighttime light (NTL) data is an emerging remote sensing source extensively employed in socio-economic research [13]. Earlier studies primarily used DMSP/OLS and NPP/VIIRS NTL data, with relatively low resolution [43]. Consequently, these data sources are limited to addressing macro-level issues such as economic development and population dynamics. The procurement of high-resolution NTL data poses significant economic and logistical challenges, resulting in fewer investigations involving high-resolution nighttime light imagery. Yangwang-1 NTL data, characterized by higher resolution, multi-band range, and low cost, can furnish significant advantages for micro-urban research, such as urban traffic evaluation. In order to compare the extraction accuracy of different NTL images, we acquired the Luojia-1 NTL image and 2020 annual NPP/VIIRS image of Wenzhou. Due to the life cycle of Luojia-1 satellite, the newest NTL data we can obtain is collected in October, 2018. The comparison of the extraction accuracy is shown in Fig 8.

The results show that road extraction through Yangwang-1 is supposed to have higher accuracy when compared with NPP/VIIRS or Luojia-1 NTL images when adopting same method. That finding is obvious and shows that NTL images with higher resolution will have higher ability to detect ground object—similar with daytime remote sensing data. We also find that the improvement of NTL resolution will increase the possibility and feasibility of using deep learning technologies—the accuracy of CA U-Net (12.64%) in VIIRS data is low than OT (17.98%) and SVM(19.53%). The potential reason may attribute to the fact that only main artery of the roads can be distinguished in low resolution images, the proportion of which is very small, that makes the training process cannot learn road information effectively if deep

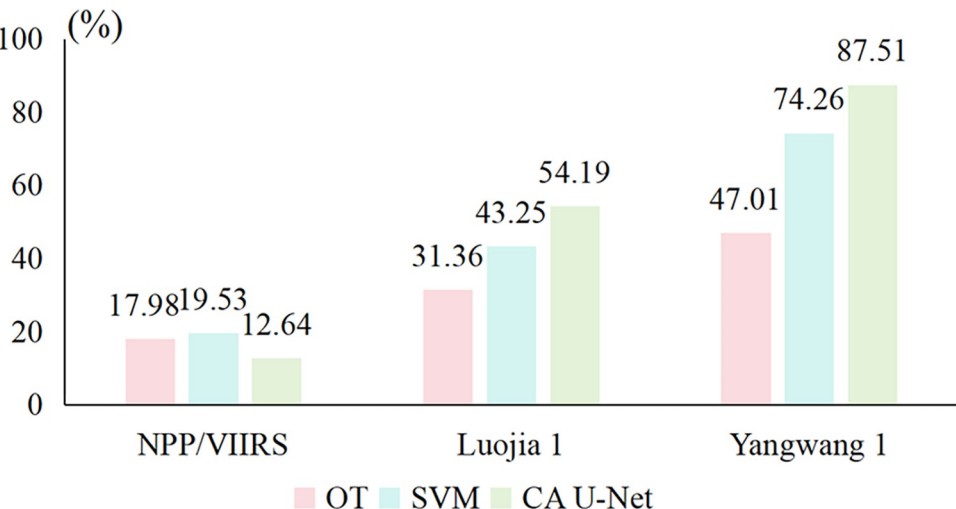

**Fig 8. Accuracy comparison between Yangwang-1 images and VIIRS, Luojia-1 images.**

learning is applied. Road extraction using NTL data remains relatively uncharted territory, and this research leverages Yangwang-1 nighttime light data to explore road extraction methodologies, thus addressing this gap in the literature. NTL primarily captures surface information under nighttime illumination, effectively circumventing the issues of discontinuous road extraction caused by tree shadows and shrubs in multi-spectral remote sensing images taken during daylight. Nonetheless, multi-spectral remote sensing data boasts superior spatial resolutions and more bands compared to nighttime light data. This study explored the potential of Yangwang-1 imagery in ground object detection through deep learning technology, which is very difficult for previous low-resolution NTL platforms.

## 4.2 Potentiality of combined framework

Within Wenzhou City, both road extraction methodologies based on Yangwang-1 NTL data yielded F1 scores above 70%, with the CA U-Net model achieving an impressive 84.46%. This highlights its efficacy for road extraction using NTL data. To validate the model's universality, we selected three representative locations in Wenzhou. Comparisons revealed that roads extracted via the CA U-Net framework more accurately corresponded to the reference road network than those extracted by SVM and OT. The SVM model exhibited satisfactory extraction accuracy in suburban areas but struggled in densely lit urban regions compared to CA U-Net. OT performed poorly across all regions, suggesting its unsuitability for standalone road extraction. However, we also realized that OT method can be combined with CA U-Net during the road extraction procedure, because the proper selection of threshold will help with the screen of background noises and blooming effects, which will decrease the accuracy of training samples. In order to test our hypothesis, we overlay the OSM roads with Yangwang-1 NTL data and find that the DN values of all the road pixels are within the range (1, 3325). According to the percentage of road pixels, we chose several ranges and use that to train the CA U-Net model. Their extraction accuracy is shown in Table 2.

From the results, we can find that the precision rate shows an increasing trend (87.51% to 92.65%) when the DN range decrease from (1,3325) to (535, 1894), and the recall rate shows a decreasing trend (81.62% to 72.76%). That means there exist a tradeoff between the selection of DV threshold and model accuracy. We can observe that the F1 score reaches highest value

**Table 2. F1 scores of CA U-Net, SVM, and OT models.**

| Lower threshold | Upper threshold | Percentage of road pixels(%) | Precision rate (%) | Recall rate (%) | F1 Score (%) |
|---|---|---|---|---|---|
| 1 | 3325 | 100 | 87.51 | 81.62 | 84.46 |
| 168 | 2689 | 95 | 90.64 | 80.54 | 85.29 |
| 426 | 2073 | 90 | 92.15 | 73.85 | 81.99 |
| 535 | 1894 | 85 | 92.65 | 72.76 | 81.51 |

85.29% when we chose the threshold range (168, 2689). The result indicates that we can improve the total model performance through proper selection of threshold if the road extraction tasks can tolerate very tiny recall losses. That also confirms our hypothesis that the traditional OT method can also be considered in future studies to improve the performance of deep learning framework.

Road network extraction continues to confront challenges demanding further exploration [14]. Since NTL data can only detect illuminated road information, its application is limited to well-lit urban roads, with low extraction accuracy in rural areas featuring dim lighting. Additionally, the high light intensity from heavily trafficked streets and the corresponding overflow phenomenon complicate road extraction using NTL data exclusively. These constraints impact the extraction of roads from NTL data. To address these issues, future research should focus on road extraction methods that integrate high-resolution multi-spectral remote sensing images captured in daylight with nighttime light images.

## 5. Conclusions

Nighttime light (NTL) remote sensing data introduces an innovative perspective to road extraction, traditionally dominated by daytime satellite imagery. This research explores the potentiality of Yangwang-1 NTL data for urban road extraction with the combination of OSM roads through a novel deep learning framework, leading to several significant conclusions.

1. Yangwang-1 data emerges as a viable source for urban road extraction. As a new generation of NTL data set in ~30 resolution, Yangwang-1 has been proven to have the potentiality to contain detailed road segmentation information through its wide DN range and good data quality. Through the experiment in Wenzhou, we find that Yangwang-1 has higher extraction accuracy in suburban and residential area and may contain a few classification errors in urban center with blooming effects and rural area with limited light intensity.

2. We propose a novel deep learning framework CA U-Net to train the NTL data with the reference of OSM roads. It achieves an F1 score of 84.46% for road extraction, surpassing SVM and OT performance, thereby underscoring the potential of deep learning frameworks in road extraction tasks using NTL data. Through the application of SVM, OT, and the CA U-Net framework for road extraction in the Wenzhou region based on NTL data, we observe F1 scores exceeding 70% for both SVM and the CA U-Net framework. This also reinforces the suitability of Yangwang-1 data for road extraction tasks.

3. We compare the Yangwang-1 extraction results with NPP/VIIRS and Luojia-1 images, the results indicate the accuracy of Yangwang-1 images is highest (87.51%). We have also discussed the potentiality of combining OT method with deep leaning framework. The results show that proper chosen of threshold will effectively screen unrelated noises or abnormal pixels, thus improving the extraction accuracy.

This research is a very preliminary exploration of Yangwang-1 NTL images in the road extraction tasks with the expectation to reduce the cost of traditional road extraction through VHR images. Therefore, there are some limitations that need to be further addressed in future studies. First, the blooming effects in city center and feeble light intensity in rural area is a major challenge for the current road extraction through NTL data. One potential solution may be the introduce of moderate-resolution daytime satellite images with low cost, like Landsat images as an information supplementation. That combination may require careful design of approaches to deal with the different data structures of them. Second, the road information is not fully made use of in our deep learning framework. We just use the OSM roads to form the road pixels and non-road pixels in the training process without using its graph information—which contain the connection information between roads. And that information will make our model have the ability to infer road segments which are partly sheltered. Finally, as we have mentioned before, there exist some noises in NTL data and OT approach can be a simple and easy way to preliminarily screen these noises before the conduction of deep learning framework. However, how to chose the best thresholds and combine deep learning technologies with traditional approaches is also a challenge job.

## Author Contributions

**Conceptualization:** Hao Zhang.

**Data curation:** Anfeng Zhu, Hao Zhang, Luyao Wang.

**Formal analysis:** Anfeng Zhu, Jun Hao.

**Funding acquisition:** Anfeng Zhu.

**Methodology:** Xu Gang, Xiaoyu Long.

**Software:** Xu Gang, Xiaoyu Long.

**Writing – review & editing:** Jun Hao.

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
