## [Decision Letter · Decision Letter 0]

16 Oct 2023

PONE-D-23-26958Urban Road Extraction Using Yangwang-1 Nighttime Light Data: An Enhanced U-Net Framework ApproachPLOS ONE

Dear Dr. Wang,

Thank you for submitting your manuscript to PLOS ONE. After careful consideration, we feel that it has merit but does not fully meet PLOS ONE’s publication criteria as it currently stands. Therefore, we invite you to submit a revised version of the manuscript that addresses the points raised during the review process.

We look forward to receiving your revised manuscript.

Kind regards,

Claudionor Ribeiro da Silva

Academic Editor

PLOS ONE

Journal Requirements:

When submitting your revision, we need you to address these additional requirements. 1. Please ensure that your manuscript meets PLOS ONE's style requirements, including those for file naming. The PLOS ONE style templates can be found athttps://journals.plos.org/plosone/s/file?id=wjVg/PLOSOne_formatting_sample_main_body.pdf andhttps://journals.plos.org/plosone/s/file?id=ba62/PLOSOne_formatting_sample_title_authors_affiliations.pdf 2. Note from Emily Chenette, Editor in Chief of PLOS ONE, and Iain Hrynaszkiewicz, Director of Open Research Solutions at PLOS: Did you know that depositing data in a repository is associated with up to a 25% citation advantage (https://doi.org/10.1371/journal.pone.0230416)? If you’ve not already done so, consider depositing your raw data in a repository to ensure your work is read, appreciated and cited by the largest possible audience. You’ll also earn an Accessible Data icon on your published paper if you deposit your data in any participating repository (https://plos.org/open-science/open-data/#accessible-data). 3. Please note that PLOS ONE has specific guidelines on code sharing for submissions in which author-generated code underpins the findings in the manuscript. In these cases, all author-generated code must be made available without restrictions upon publication of the work. Please review our guidelines at https://journals.plos.org/plosone/s/materials-and-software-sharing#loc-sharing-code and ensure that your code is shared in a way that follows best practice and facilitates reproducibility and reuse. 4. Thank you for stating the following financial disclosure: "This research was funded by Basic Social Development Science and Technology Project of Wen- 373zhou Science and Technology Bureau（Grant NO.S20220017）" Please state what role the funders took in the study.  If the funders had no role, please state: ""The funders had no role in study design, data collection and analysis, decision to publish, or preparation of the manuscript."" If this statement is not correct you must amend it as needed. Please include this amended Role of Funder statement in your cover letter; we will change the online submission form on your behalf. 5. We note that Figures 3 to 5 in your submission contain map/satellite images which may be copyrighted. All PLOS content is published under the Creative Commons Attribution License (CC BY 4.0), which means that the manuscript, images, and Supporting Information files will be freely available online, and any third party is permitted to access, download, copy, distribute, and use these materials in any way, even commercially, with proper attribution. For these reasons, we cannot publish previously copyrighted maps or satellite images created using proprietary data, such as Google software (Google Maps, Street View, and Earth). For more information, see our copyright guidelines: http://journals.plos.org/plosone/s/licenses-and-copyright. We require you to either (1) present written permission from the copyright holder to publish these figures specifically under the CC BY 4.0 license, or (2) remove the figures from your submission: 1.) You may seek permission from the original copyright holder of Figures 3 to 5 to publish the content specifically under the CC BY 4.0 license.   We recommend that you contact the original copyright holder with the Content Permission Form (http://journals.plos.org/plosone/s/file?id=7c09/content-permission-form.pdf) and the following text:“I request permission for the open-access journal PLOS ONE to publish XXX under the Creative Commons Attribution License (CCAL) CC BY 4.0 (http://creativecommons.org/licenses/by/4.0/). Please be aware that this license allows unrestricted use and distribution, even commercially, by third parties. Please reply and provide explicit written permission to publish XXX under a CC BY license and complete the attached form.” Please upload the completed Content Permission Form or other proof of granted permissions as an ""Other"" file with your submission. In the figure caption of the copyrighted figure, please include the following text: “Reprinted from [ref] under a CC BY license, with permission from [name of publisher], original copyright [original copyright year].” 2.) If you are unable to obtain permission from the original copyright holder to publish these figures under the CC BY 4.0 license or if the copyright holder’s requirements are incompatible with the CC BY 4.0 license, please either i) remove the figure or ii) supply a replacement figure that complies with the CC BY 4.0 license. Please check copyright information on all replacement figures and update the figure caption with source information. If applicable, please specify in the figure caption text when a figure is similar but not identical to the original image and is therefore for illustrative purposes only. The following resources for replacing copyrighted map figures may be helpful: USGS National Map Viewer (public domain): http://viewer.nationalmap.gov/viewer/The Gateway to Astronaut Photography of Earth (public domain): http://eol.jsc.nasa.gov/sseop/clickmap/Maps at the CIA (public domain): https://www.cia.gov/library/publications/the-world-factbook/index.html and https://www.cia.gov/library/publications/cia-maps-publications/index.htmlNASA Earth Observatory (public domain): http://earthobservatory.nasa.gov/Landsat: http://landsat.visibleearth.nasa.gov/USGS EROS (Earth Resources Observatory and Science (EROS) Center) (public domain): http://eros.usgs.gov/#Natural Earth (public domain): http://www.naturalearthdata.com/

**Additional Editor Comments:**

The manuscript must be corrected in all points indicated by the reviewers, such as:

1. The Title is doesn’t align with the objectives and findings of the research.

2. The novelty of this paper is not clear. The difference between present work and previous Works should be highlighted. Also, needs to include similar previous studies and discuss them.

3. Data preparation and preprocessing is not well explained.

4. Generalized research method workflow is missing.

5. Is the method used the same as that presented in reference nº 30? If it exists, write clearly what kind of improvement is added in this method.

6. Software and/programs used in the research are not included.

7. The paper lacks a thorough comparison of the cited works and a detailed discussion of the findings.

8. When comparing methods, use accuracy and write clearly, which platform, what software, and how long time for the different method.

9. Conclusions is more of an afterthought. The authors are suggested to include afterthought of this work.

10. Recommend some avenues for future research at the end of the conclusion section.

Reviewers' comments:

Reviewer's Responses to Questions

**Comments to the Author**

1. Is the manuscript technically sound, and do the data support the conclusions?

Reviewer #1: Yes

Reviewer #2: Partly

2. Has the statistical analysis been performed appropriately and rigorously? 

Reviewer #1: Yes

Reviewer #2: No

3. Have the authors made all data underlying the findings in their manuscript fully available?

Reviewer #1: Yes

Reviewer #2: No

4. Is the manuscript presented in an intelligible fashion and written in standard English?

Reviewer #1: No

Reviewer #2: No

5. Review Comments to the Author

Reviewer #1: 1. This paper use Yangwang-1 Nighttime Light Data with 38m resolution to extract road, it is a new data, so I would like to introduce it to publish.

2. The author said: In this study, we adopt the U-Net framework, enhanced with a Cross-Attention Module

(CAM), for road extraction from NTL data[30]. It looks they used a method which have been published in reference No.30.

So does that mean this method has been published somewhere and here there is no innovation in method? if not, please write clearly what kind of improvement is added in this paper.

3. To compare the method performance, accuracy is one aspect, the author should also write clearly, which platform, what software, and how long time for the different method.

Reviewer #2: It has been indicated in the attachment.

1. The Title is doesn’t align with the objectives and findings of the research.

2. The novelty of this paper is not clear. The difference between present work and previous Works should be

highlighted. Also, needs to include similar previous studies and discuss them.

3. Section 3.1 should be relocated to Section 2

4. Data preparation and preprocessing is not well explained.

5. Generalized research method workflow is missing

6. Software and/programs used in the research are not included

7. The paper lacks a thorough comparison of the cited works and a detailed discussion of the findings.

8. CONCLUSIONS is more of an afterthought. The authors are suggested to include afterthought of this work.

9. Recommend some avenues for future research at the end of the conclusion section

6. PLOS authors have the option to publish the peer review history of their article (what does this mean?). If published, this will include your full peer review and any attached files.

Reviewer #1: No

Reviewer #2: **Yes: **Seyoum Eshetie

---

## [Author Response · Author response to Decision Letter 0]

2 Dec 2023

Thanks for the reviewers helpful comments on my manuscript “Road Extraction Through Yangwang-1 Nighttime Light Data: A Case Study in Wenzhou, China”. I appreciate the valuable feedback provided, and I have addressed all the comments and suggestions to improve the quality of the manuscript. The detailed responses are attached in the .docx files.

---

## [Decision Letter · Decision Letter 1]

29 Dec 2023

Road Extraction Through Yangwang-1 Nighttime Light Data: A Case Study in Wenzhou, China

PONE-D-23-26958R1

Dear Dr. Wang,

We’re pleased to inform you that your manuscript has been judged scientifically suitable for publication and will be formally accepted for publication once it meets all outstanding technical requirements.

Kind regards,

Claudionor Ribeiro da Silva

Academic Editor

PLOS ONE

Additional Editor Comments (optional):

Reviewers' comments:

Reviewer's Responses to Questions

**Comments to the Author**

1. If the authors have adequately addressed your comments raised in a previous round of review and you feel that this manuscript is now acceptable for publication, you may indicate that here to bypass the “Comments to the Author” section, enter your conflict of interest statement in the “Confidential to Editor” section, and submit your "Accept" recommendation.

Reviewer #1: All comments have been addressed

Reviewer #2: All comments have been addressed

2. Is the manuscript technically sound, and do the data support the conclusions?

Reviewer #1: Yes

Reviewer #2: Yes

3. Has the statistical analysis been performed appropriately and rigorously? 

Reviewer #1: Yes

Reviewer #2: Yes

4. Have the authors made all data underlying the findings in their manuscript fully available?

Reviewer #1: Yes

Reviewer #2: No

5. Is the manuscript presented in an intelligible fashion and written in standard English?

Reviewer #1: Yes

Reviewer #2: Yes

6. Review Comments to the Author

Reviewer #1: I think the authors have answered the questions in the last review satisfactory, so I agree to publish.

Reviewer #2: 1.The paper needs a thorough comparison of the findings with other works in the Discussion section

The Discussion section, which follows the Results section, is where researchers interpret and explain the meaning of the results in the context of the research question and existing literature. In this section, researchers connect their findings to previous studies, make explicit connections back to their research question(s), and provide an explanation of how the results might be generalized. It is in the Discussion section that researchers answer their research questions and compare their findings with those of other scholars.

7. PLOS authors have the option to publish the peer review history of their article (what does this mean?). If published, this will include your full peer review and any attached files.

Reviewer #1: No

Reviewer #2: **Yes: **Seyoum Melese Eshetie

---

## [Editor Report · Acceptance letter]

11 Jan 2024

PONE-D-23-26958R1 

PLOS ONE

Dear Dr. Wang, 

I'm pleased to inform you that your manuscript has been deemed suitable for publication in PLOS ONE. Congratulations! Your manuscript is now being handed over to our production team.

Kind regards, 

on behalf of

Dr. Claudionor Ribeiro da Silva 

Academic Editor

PLOS ONE